# Psychosocial Occupational Health—A Priority for Middle-Income Countries?

**DOI:** 10.3390/healthcare11222988

**Published:** 2023-11-19

**Authors:** Johannes Siegrist

**Affiliations:** Centre for Health and Society, Faculty of Medicine, Heinrich-Heine-University Düsseldorf, 40225 Düsseldorf, Germany; siegrist@uni-duesseldorf.de

**Keywords:** middle-income countries, economic globalisation, psychosocial work environments, occupational health, Asia Pacific, Latin America

## Abstract

In response to new developments of work and employment in high-income countries (HICs), psychosocial aspects of work and health have received increased attention. In contrast, middle-income countries (MICs) are mainly concerned with severe challenges of noxious- and dangerous-material work environments, poor employment conditions, and deficient social policies, which leaves the psychosocial aspects with a marginal role, at best, in occupational health. More recently, differences between these two worlds were even aggravated by the COVID-19 pandemic. Yet, with economic globalisation and the growing worldwide interconnectivity, the world of work in MICs is being rapidly transformed, starting to share several concerns with the modern Western societies. In this process, psychosocial occupational health will become an increasingly pressing issue. This contribution explores the extent to which psychosocial aspects of work and health are already addressed in research originating from MICs. Using a narrative review approach, a selective focus on recent findings from two regions, Asia Pacific and Latin America, revealed an increasing interest in work stress-related problems, but a restricted impact of the respective research findings. It is hoped that future scientific developments in MICs will enrich the international state of the art in this field.

## 1. Introduction

The world of work and employment underwent substantial changes during the past few decades. Economic globalisation, a process of increasing transnational trade liberalisation and market expansion in developing countries, and the availability of ground-breaking innovations in information and communication technology were—and continue to be—two major drivers of this change. Transnational flows of goods and commodities, capital, and human labour promoted an unprecedented global interconnectivity. In high-income countries (HICs), new developments of work became evident, such as a rapid growth of service, information, and communication occupations and professions in conjunction with a decrease in the industrial sector, a widespread technological transformation of jobs, and an augmentation in the flexibility, diversity, and insecurity of employment conditions, including the growth of nonstandard work. If compared to the industrial stage of development, the main challenges of occupational safety and health in these countries shifted from a main concern about noxious-material (chemical, physical, biological) work environments and occupational injuries to a main concern about psycho-mental and socio-emotional stressors at work, evolving from work intensification, mental challenges, interpersonal conflicts, threatening employment conditions, and the required rapid adaptation to changing job tasks and job trajectories [1]. Accordingly, at the level of scientific analysis, the scope was extended beyond the traditional disciplines of occupational medicine and safety sciences to include psychological, social, behavioural, and organisational aspects, promoting occupational health science as an inter-disciplinary approach [2,3,4]. To date, a solid body of knowledge on new occupational risk and protective factors is available, and many HICs have extended their monitoring and prevention activities at work to address these new challenges [5].

In a sharp contrast to this development, occupational safety and health in most middle-income countries (MICs) is faced with substantial difficulties in tackling noxious-material work environments and occupational injuries in the agricultural and industrial sectors of productivity, in improving access to paid work and reducing informal employment, and in providing basic social protection, including health care, rehabilitation, unemployment benefits, and pensions. Moreover, universal human rights and antidiscrimination policies need to be enforced at work [6,7]. Against these far-reaching threats resulting in a high burden of work-related diseases and human suffering, any reference to problems related to psychosocial occupational health seem to be misplaced. Yet, despite substantial differences in working conditions and health between countries in the Global North and rapidly developing countries in the Global South, the world of work in MICs is being rapidly transformed through economic globalisation, technological innovations, and increased worldwide interconnectivity, starting to share several concerns with modern Western societies. The challenges of psychosocial occupational health are considered one such shared concern, given the globally experienced threats of job insecurity, often combined with a growing work intensity, and the rapid pace of change of work demands and work environments. Against this background, it is of interest to examine the extent to which scientific research originating from MICs has already addressed these latter challenges. Using a narrative literature review, this contribution explores the evolution of psychosocial occupational health research in the recent past in two regions of MICs, Latin America and Asia Pacific. As occupational health research is quite developed in several countries within these regions, there is reason to believe that the newly occurring problems of stressful psychosocial work have been tackled to some extent. As background information to this exploration, it is instructive to briefly consider how the link between work and health in MICs differs from the one in the Global North.

### Work and Health in Middle-Income Countries: Differences and Assimilation to the Global North

A few years before the outbreak of the new pandemic, the United Nations’ Sustainable Development Goals (SDGs) were declared and subsequently adopted by a large number of member states. One of these goals, SDG 8, calls for the promotion of productive employment and decent work [8]. At that time, it was evident that the working and employment conditions differed substantially between the regions of the world, most obviously between high-income, middle-income, and low-income countries. In its World Employment Social Outlook in 2018, the International Labour Organisation (ILO) estimated that every third worker in developing countries was living in extreme poverty, and many more were in vulnerable employment (own-account workers and contributing family workers) [9]. Although extreme poverty appeared significantly reduced in emerging countries, the pace of reduction had slowed down, and more than 400 million people were left in moderate working poverty in developing and emerging countries [9]. Informal employment remained extremely high in low-income countries, with less than twenty percent of workers being formally employed, as compared to some sixty percent in upper-middle-income countries, and some eighty-seven percent in high-income countries [10]. Unemployment and under-employment continued to be main concerns in the Global South, in concert with massive restrictions of social protection and civil rights.

While far-reaching decent work deficits were substantial in these parts of the world, MICs experienced some progress. Overall, they showed a strong increase in gross domestic product over the past twenty-five years. Six countries stand out in this regard: Brazil, China, India, Indonesia, Mexico, and the Russian Federation [11]. Despite national variations, these countries underwent a process of structural transformation with three dominant features. First, the proportion of people employed in agriculture declined quite strongly. Compared to the year 1991, the percentage in these countries was reduced from fifty-one percent to twenty-seven percent by 2017 [11]. At the same time, as a second feature, the industrial sector did not grow as expected from previous developments in HICs. Overall, about a quarter of the workforce was employed in industry, but many jobs remained of low quality. In the context of economic globalisation, industrial productivity was dominated by multinational corporations operating from the Global North, such that the contribution of the manufacturing sector to the economic progress within MICs remained constrained. Third, during this time period, a rapid increase in the service sector was observed, absorbing currently about half the employed population of these six countries. The main problem of this latter development consists in the fact that employment growth was largely limited to low-productivity service sectors, such as retail trade or low-skilled information and communication jobs. These jobs offer poor and insecure working conditions and low pay [11]. In consequence, the level of precarious work in MICs is still high, with more than forty percent of workers considered as being in vulnerable or informal employment. However, while India and Indonesia have documented the highest proportions of precarious work, the trends in Russia, Mexico, and Brazil are more favourable, and China has witnessed the sharpest decrease in vulnerable employment over time [11]. These difficulties and disadvantages among working people in MICs were aggravated by the COVID-19 pandemic. Compared to the workforce in HICs, rates of infection, severity of disease, and fatality rates were substantially higher, not least due to lack of vaccination and restricted access to hospital treatment. Moreover, lockdown measures, job loss, and lack of financial and social support from national policies exacerbated human suffering and triggered a mental health crisis [12,13]. The pandemic widened social inequalities in health that dominated the spectrum of morbidity and mortality [14].

However, there are also some promising developments in emerging economies. The young generations entering the labour market are much better educated than the previous generations, offering a broad spectrum of technological skills and competences. For instance, in upper-middle-income countries, only some thirty percent of the workforce are estimated to be undereducated with regard to the demands of the labour market (and some twenty percent are estimated to be overeducated), whereas the percentage of undereducated amounts to seventy percent in low-income countries [15]. Skill match is an important driver of productivity gain and economic growth. Not least in the reaction to the lockdown measures, the COVID-19 pandemic has accelerated the implementation of technological innovations in these regions. Remote work, digital work, rapid automation, and e-commerce suddenly expanded. Delivery services via digital platforms, such as on-demand physical services or crowd work, are a prominent example. Moreover, the pandemic has accelerated the automation of tasks in a large number of globally operating businesses [16]. As a further side effect of economic globalisation, a considerable part of workers in MICs are employed in the global supply chain economy with strong linkages to HICs. Through these linkages, employment regulations were improved, securing wages and basic safety protection.

While the two regions of Latin America and Asia Pacific differ quite markedly in terms of sociocultural and sociopolitical development, the countries within each region exhibit large variations as well. In Asia Pacific, a region with more than a dozen of nations, three countries—Japan, Australia, and South Korea—are classified as high-income nations, whereas China, Malaysia, and Thailand belong to the upper-middle-income country (UMIC) category, and Vietnam and Indonesia are lower-middle-income countries (LMICs). In Latin America, Brazil is the largest UMIC, and Mexico, Costa Rica, and Chile are members of the Organisation for Economic Co-Operation and Development (OECD). The global financial crisis of 2008 decelerated the economic growth at large scale, thus reducing several promising developments of decent work and employment, leaving informal and precarious work as main threats across this region [17,18]. Yet, more recently, several countries within Latin America improved their occupational health and safety services, implementing laws and regulations to protect workers’ health. Colombia and Mexico are the best examples, but similar initiatives were developed in Argentina, Chile, Uruguay, Puerto Rico, Brazil, Venezuela, and Peru [18]. In the Asia Pacific region, HICs stand out with their well-developed occupational health services and policies. However, large deficits in occupational health protection are obvious in the remaining countries, most obviously in Indonesia and Vietnam [19,20]. To conclude, MICs still face substantial challenges to meet the UN goal of promoting productive employment and decent work. Updated scientific evidence on the adverse health effects of disadvantaged physical and psychosocial work environments represents an important step towards promoting this goal. It is therefore of interest to explore the current state of this evidence, specifically with respect to the psychosocial work environment.

## 2. Methods

Narrative reviews aim at identifying specific directions of research without following a standardized methodology of literature search. They provide a first overview of a field of research as a basis for extended, more systematic inquiry. Often based on available systematic reviews or books, enriched by online searches and contacts with relevant research networks, they serve as orientation for main directions, challenges, and open questions [21]. This approach was chosen here as an entry point to a field of research that requires much more resources and time than available but deserves a preliminary assessment due to its current topicality. In the case of Latin America, a recent systematic review summarizing research on psychosocial work environment and health from 2010 to 2021 is available, providing insights based on 85 research reports [22]. This review was complemented by a thematic online search of relevant journals for the period from January 2020 to September 2023. The following journals with English language papers were retrieved from references in this review: American Journal of Industrial Medicine; International Archives of Occupational and Environmental Health; Industrial Health; Stress and Health; and Safety and Health at Work (Title and Abstract screening). Additionally, an update was performed by expert consultation. Concerning the Asia Pacific region, this review approach was more difficult, as research developments differ substantially between high-income and middle-income countries. Moreover, no updated topical systematic review was available. Here, the narrative review was based on two books summarizing scientific evidence on psychosocial work and health, published in 2014 and 2016 [23,24] and on a thematic online search of relevant journals for the period from January 2017 to September 2023. These journals were Asia Pacific Journal of Public Health; Industrial Health; International Archives of Occupational and Environmental Health; Journal of Occupational Health; and Safety and Health at Work (Title and Abstract screening). Moreover, the website of the Asia Pacific Academy for Psychosocial Factors at Work was retrieved, together with an overview of recent conference contributions. As a result, the scope of analysis was considerably extended, but remained substantially selective due to the limitations of the search procedure.

## 3. Results

### 3.1. Latin America

For a long time, there was a paucity of data on the working conditions and health in the countries of this region. Yet, some years ago, a first cross-country comparative survey on working conditions and health was accomplished in ten Latin American countries [25]. The samples were restricted to formally employed men and women working in the industry, construction, and service sectors. Agricultural work and mining were excluded. The survey listed several physical, chemical, and biological hazards. In addition, lack of influence at work (order of tasks, choice of working methods) and working at high speed were included as psychosocial risk factors. Findings from Colombia, Argentina, Chile, six countries of Central America, and Uruguay revealed a high percentage of employees working more than 40 h/week (58 percent among men; 40 percent among women), a high prevalence of repetitive movements, manual handling, as well as working at high speed and with low influence [25]. Beginning in 2010, a couple of studies were conducted that mainly examined two established theoretical models of stressful psychosocial work environments, i.e., demand–control–support [26] and effort–reward imbalance [27]. As these models were developed and tested during the last quarter of the 20th century in Europe and in the USA, it was obvious to explore their applicability and merit in the Latin American context. These early studies were part of a carefully conducted systematic review of publications on psychosocial work and health, covering the period from 2010 to 2021 [22]. This highly informative review, based, as mentioned, on 85 studies, revealed a profile of research that was characterized by the following features. Most studies were based on a cross-sectional design and included small samples. In the majority of cases, health measures were restricted to self-reported data, with depressive symptoms and musculoskeletal pain being studied most often. About half of all studies were conducted among health care professionals or among educational personnel. While a larger part of the studies focused on health indicators, several investigations analysed behavioural measures, such as work ability, absenteeism, intention to leave, or accidents. Of interest, more than half of all studies were conducted in Brazil. Overall, a body of new knowledge resulted from these studies, indicating that work defined by high demands, low control, and low support and employment conditions characterized by an imbalance between high effort spent and low reward received in turn are associated with poor health and poor functioning. Moreover, some additional work-related aspects were included, such as mobbing and violence or poor leadership style. However, the quality of this knowledge was rather modest, reflecting restricted funding and training opportunities and structural constraints of the academic work. In consequence, its impact in terms of science and policy was limited [22].

To update the state of the art in this field, the journals mentioned were screened from January 2000 to September 2023. In a dozen of papers identified in this review, several conceptual and methodological improvements were observed, and new health indicators were included. Conceptually, one publication analysed different combinations of the components of the demand–control–support model in relation to health [28], while two further papers tested the effort–reward imbalance model in association with depression, performing interaction analysis and exploring gender-specific effects [29,30]. In terms of methods, longitudinal designs [31,32] and multilevel analyses [33] were performed. Cognitive function [34], prediabetes [35], metabolic syndrome [36], and mental health in relation to COVID-19 [37] were studied as novel health indicators.

In conclusion, in several Latin American countries, and most strongly in Brazil, research on psychosocial work environment and health has steadily grown over the past ten years, improving its scientific quality and moving from a local or national level to international visibility and collaborations. If continued and structurally supported, this research will narrow the developmental gap in science between high-income and middle-income countries.

### 3.2. Asia Pacific

Research on psychosocial occupational health emerged quite differently across this region. While Japan [38] and Australia [39] initiated internationally acknowledged research traditions already in the 1990s, Chinese scientists introduced the measurement methods of leading theoretical models of psychosocial work environments (e.g., demand–control–support; effort–reward imbalance) in their country at the beginning of this century [40]. A cross-national Asia Pacific research effort started in 2010 with a series of annual expert workshops that resulted in innovative transnational scientific collaborations and in the production of two books reviewing emerging original research in this region, specifically, in Australia, Japan, South Korea, Malaysia, Thailand, China, Indonesia, and Vietnam [23,24].

These two books are particularly remarkable as they not only document the adaptation of Western research to these rapidly developing countries, but also emphasise specific socio-cultural features and contexts of Asian working life, thus enriching the global scope of analysis. For instance, bullying as a psychosocial risk factor is generally found to be an unacceptable offensive behaviour in Western societies, whereas in a collectivist culture like Malaysia that expects a high degree of loyalty to superiors, bullying is experienced rather frequently, specifically among low-status and newly recruited employees, targeted by seniors as a sign of unequal power [41]. The gender-role-specific impact of the conflict between work and family (and family and work) on health and well-being in China is another example. Compared to Western societies, the gendered division of work is more pervasive, exposing a large part of fulltime-employed women to a double burden of paid work and unpaid family work [42]. In consequence, women exposed to cumulative stress are particularly vulnerable to cardiovascular risk and disease [43]. In some Asian countries, the working and employment conditions are hardly comparable to those in Western HICs. As an example, in Thailand, Indonesia, Vietnam, and Malaysia, retail work is highly prevalent. Up to ninety percent of employed people work in small or medium enterprises, where critical ergonomic conditions, highly repetitive movements, and work pressure, combined with job insecurity and low pay, determine the everyday work experience [44]. Finally, labour and social policies differ substantially across Asian–Pacific countries. A comparison of these measures between two high-income regions, Australia and Taiwan, and two middle-income countries, Malaysia and Thailand, illustrates the case. Concerning laws on workers’ mental health and safety, policies in the former two regions define clear employer responsibilities by specific laws, and some disorders (e.g., cardiovascular diseases in Taiwan) are recognised as compensable occupational diseases in relation to psychosocial stress at work. Neither Malaysia nor Thailand have implemented binding obligations beyond very general statements. If specified, occupational safety and health measures are restricted to physical and chemical hazards [45].

While the two volumes give an impressive account of well-developed and innovative research on psychosocial stress at work and health in Asia Pacific, its dynamic growth becomes only apparent if the most recent scientific productivity is reviewed. In the narrative review of papers published in the five selected journals between 2017 and 2023, more than fifty publications dealing with psychosocial occupational health were identified, with multiple contributions from South Korea, Japan, and Australia. It is not possible to list them all here, but three pioneering features deserve to be highlighted. First, several investigations addressed updated ways of monitoring psychosocial risks and implementing worksite health promotion. They mainly focused on participatory approaches and used novel biomarkers for evaluation [46,47,48,49]. A second line of contributions analysed the health effects of non-standard employment conditions evolving from new global labour market developments. Unpredictable work trajectories and increased job instability had a strong negative impact on mental and physical well-being [50,51,52,53]. Finally, as long working hours are a highly prevalent concern in Asia Pacific, their adverse effects on a broad spectrum of health complaints were extensively studied, often as part of national working conditions surveys [54,55,56,57]. The bulk of this evidence resulted from studies performed in Japan, South Korea, and Japan. Yet, among MICs, China stands out with rapidly growing scientific productivity in this field [58]. Of interest, Chinese researchers contributed internationally, leading analyses on the interplay of adverse working conditions with infection risks due to the COVID-19 pandemic [59,60].

To sum up, during most recent years, psychosocial occupational health research in Asia Pacific has grown to a mature field of enquiry, thus minimizing its distance to the scientific developments in high-income countries.

## 4. Discussion

This selective review of recent research on psychosocial occupational risks and health in two regions of MICs, Latin America and Asia Pacific, revealed a high degree of awareness of, and inquiry into, work stress-related problems within these countries. Importantly, in addition to incorporating research progress from HICs, these analyses identified new aspects within specific cultural and socio-economic environments, specifically in the context of Asian research. In summary, in both regions, psychosocial occupational research has grown very rapidly in recent years and has received wide attention within scientific audiences as well as among stakeholders of occupational health (professionals, employers, trade unions). In Latin America, knowledge, so far, is restricted to more privileged occupational groups (formal employment in the service sector and public institutions), and the quality of evidence is limited due to the predominance of cross-sectional study designs. These methodological limitations are mainly due to restricted funding and training opportunities, as well as to the structural constraints of academic research. While these latter concerns matter for developing countries in Asia Pacific as well, differences in research development are more pronounced between high- and middle-income countries. Thanks to a transnational collaborative development inaugurated by Australian and Japanese scientists [23,24], a series of cross-national investigations was initiated, thus promoting the field in less developed regions.

The findings of these reports do not support the notion that occupational health research in MICs is mainly concerned with occupational injuries, industrial hygiene, and traditional occupational medicine, dealing with physical and chemical hazards. Rather, occupational health researchers demonstrated a high degree of awareness of new challenges resulting from the transformation of work and employment in a globalised economy. In pioneering studies, they set out to document the prevalence of psychosocial stress at work in their country and the burden of disease that may be attributable to it. The findings of many relevant investigations were not published in English language, but rather in the original language of the respective countries, mainly Spanish or Portuguese in Latin America, and Chinese or Japanese in Asia. For instance, in the Latin American review highlighted above, it was stated that most accounts of research in this region “considered only papers in English, indexed in international databases, … thus ignoring to a considerable extent studies conducted in Latin America” ([22], p. 188). These signs of a hegemonial role of Western science have contributed to a lack of awareness and recognition of original research evolving in rapidly developing regions of the world.

In this perspective, the main difference in occupational health research between the Global North and the Global South is not the presence or absence of psychosocial occupational health research, but rather the transfer of respective knowledge into practice and policy. The documents confirmed that laws and regulations protecting workers’ health and safety and securing decent work are insufficiently developed and enforced in rapidly developing countries, and that national labour and social policies often fail to provide basic protection and support to employed people. As a further shortcoming, occupational safety and health services remain underdeveloped, with a substantial shortage of health professionals to be recruited from different disciplines and training backgrounds. In both regions considered, the impact of trade unions on the promotion of healthy work is limited, as they represent only a part of the employed population. Few if any MICs in the two regions collect administrative monitoring and surveillance data on occupational health on a regular basis, and opportunities of developing preventive measures from available evidence are rare. Taken together, there is still a long way to go before innovative research on psychosocial occupational health in these developing countries can be transferred into marked and sustainable effects on workers’ health.

This short report suffers from several limitations. Importantly, it is based on selective research, as no attempt was made to review a broad field of scientific inquiry in a systematic way. Its main aim was to challenge the widely prevalent view that occupational health research in MICs mainly deals with the traditional topics of safety, hygiene, and material hazards, thus bypassing new evidence related to psychosocial occupational health emerging from research in modern Western societies. A second limitation concerns the lack of an in-depth analysis of the findings across different regions or different socioeconomic contexts. No direct access to data, as a prerequisite of conducting comparative analyses, was available. Third, given the restriction to two regions, the findings of this study cannot be generalized to other regions of the world. However, it should be noticed that Latin America and Asia Pacific stand out as those regions within the Global South that document the most productive scientific achievements in this field.

## 5. Conclusions

This report demonstrated significant advances in research on psychosocial occupational health in two broad regions of middle-income countries, Latin America and Asia Pacific. Despite the prevalence of traditional occupational hazards and disadvantaged employment conditions, new challenges of a globally transformed world of work are increasingly addressed by scientists in these regions. Thus, the developmental gap with respect to a more advanced research tradition in high-income countries is diminishing. Yet, large efforts are needed to transfer this new evidence into marked and sustainable effects on workers’ health.

## Data Availability

Reviewed publications can eventually be requested from the author.

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
