# Peer review of "Psychosocial Occupational Health—A Priority for Middle-Income Countries?"

_healthcare, 2023, doi:10.3390/healthcare11222988_

Round 1

Reviewer 1 Report

Comments and Suggestions for Authors
  • In the Introduction, consider briefly mentioning the scope and methods of the literature review, i.e., that it is a selective narrative review focused on Latin America and Asia Pacific regions.
  • In the Asia Pacific section, highlight some key findings from the cross-national collaborations and books you mention, to give more insight into the regional research.
  • In the Discussion, comment on whether your main conclusions may also apply to other regions like Africa, Central Asia, etc. or if more research is needed.
  • Consider reducing the length of some sentences for easier readability.
Comments on the Quality of English Language
  • Overall, the paper is well-written in clear academic English.
  • There are some long, complex sentences that could be simplified by breaking them up into shorter sentences. This would enhance readability.

Author Response

Thank you very much for your constructive comments. The manuscript has now been substantially revised and improved, following the established procedure (Introduction; Methods; Results; Discussion; Conclusions). My answers to your comments are as follows:

  1. Introduction: Given a rather broad framework of this manuscript, the Introduction starts by elaborating some key differences of working conditions and of occupational health practice and research between high-income and middle-income countries. Given this background, the scope and method of this contribution is described as follows (line 54ff.):

“Against this background, it is of interest to examine to what extent scientific research originating from MICs has already addressed these latter challenges. Using a narrative literature review, this contribution explores the evolution of psychosocial occupational health research in recent past in two regions of MICs, Latin America and Asia Pacific. As occupational health research is quite developed in several countries within these regions there is reason to believe that the newly occurring problems of stressful psychosocial work have been tackled to some extent.”

  1. As mentioned in my response to Reviewer 2, a new Methods section was introduced to describe the procedure of data collection and analysis. Concerning the section on Asia Pacific, a thematic online search of relevant journals for the period from January 2017 to September 2023 was included to the description of the two volumes, and highlights of recent research were described (see text line 282-301).
  1. In the Discussion section, study limitations are emphasized more clearly, including the limitation of generalizing the insights of this analysis to other regions (see line 356ff.).
  1. Several sentences were shortened throughout the text to improve the reading.

Reviewer 2 Report

Comments and Suggestions for Authors

My verdict as reviewer of this manuscript is NOT to accept it.

I indicate the reasons that have led me to this decision:

- Introduction: It is not expressly stated:

What is the main novelty of this work, what are its main contributions to the research area addressed, and what are its main research proposals? What are its research proposals?

- Methodology: Completely absent. There is no section on the methodology used to reproduce the study. You should provide enough detail to allow your work to be reproduced. If it is an already published method it should be indicated by a reference where only relevant modifications should be described.

If, as you indicate, this is a review study, you do not make it clear which literature search method was used, which databases were used, search criteria, inclusion or rejection criteria and the results obtained from the review.

- Conclusions: Absence of a section on the conclusions reached in your study. Similarly, it should be considered to describe the limitations you have had in carrying out the work and indicate future lines of research

Author Response

In response to your critique, the manuscript was substantially revised and improved, following the established procedure (Introduction; Methods; Results; Discussion; Conclusions). My answers to your comments are as follows:

  1. The novelty of this contribution consists in raising awareness among scientists and other stakeholders about a challenging research gap in middle-income countries: the analysis of expanding stressful psychosocial work environments and their adverse effects on workers’ health. This challenge is elaborated by a narrative review of current research in two regions, Asia Pacific and Latin America.

The scope and method of this contribution is described in the Introduction as follows (line 54ff.):

“Against this background, it is of interest to examine to what extent scientific research originating from MICs has already addressed these latter challenges. Using a narrative literature review, this contribution explores the evolution of psychosocial occupational health research in recent past in two regions of MICs, Latin America and Asia Pacific. As occupational health research is quite developed in several countries within these regions there is reason to believe that the newly occurring problems of stressful psychosocial work have been tackled to some extent.”

  1. In the newly included Methods section, the narrative literature review is described, with journals and time frame of screening mentioned in detail. In fact, an extended literature search was conducted to improve the empirical basis of this contribution. The respective section is as follows (line 171ff.):

“In case of Latin America, a recent systematic review summarizing research on psychosocial work environment and health from 2010 to 2021 was available, providing insights based on 85 research reports [22]. This review was complemented by thematic online search of relevant journals for the period from January 2020 to September 2023. The following journals with English language papers were retrieved from references in this review: American Journal of Industrial Medicine; International Archives of Occupational and Environmental Health; Industrial Health; Stress and Health; Safety and Health at Work (Title and Abstract screening). Additionally, an update occurred by expert consultation. Concerning the Asian Pacific region, this review approach was more difficult as research developments differ substantially between high- income and middle- income countries. Moreover, no updated topical systematic review was available. Here, the narrative review was based on two books summarizing scientific evidence on psychosocial work and health, published in 2014 and 2016 [23, 24] and on a thematic online search of relevant journals for the period from January 2017 to September 2023. These journals were Asia Pacific Journal of Public Health; Industrial Health; International Archives of Occupational and Environmental Health; Journal of Occupational Health; Safety and Health at Work (Title and Abstract screening). Moreover, the website of the Asian Pacific Academy for Psychosocial Factors at Work was retrieved, together with an overview of recent conference contributions”.

  1. The Discussion section was revised and included a section on study limitations including the limitation of generalizing the insights of this analysis to other regions (see line 356ff.).

A section Conclusions was added to the manuscript (see line 369ff.).. 

Reviewer 3 Report

Comments and Suggestions for Authors

Dear authors,

Thank you for the opportunity to review your review work, which addresses important issues in psychosocial occupational health in Latin America and the Asia-Pacific region.

A detailed acquaintance with your work allowed us to highlight a number of recommendations:

Strengthen the content of the work with a large analysis of scientific articles (the total number of sources used is 43, which is relatively small for a review type article).

In the main part of the article, I would like to see which documents and articles were selected by the author for this review, what the author focused on, and which works were ignored. This would help to better understand the author's conclusions and assumptions.

In discussing the results, I would like to see more clearly the existing problems and possible directions for further research.

The article does not have a Conclusion section. It needs to be added.

Best wishes, reviewer

Author Response

Thank you very much for your constructive comments. The manuscript has now been substantially revised and improved, following the established procedure (Introduction; Methods; Results; Discussion; Conclusions). My answers to your comments are as follows:

  1. In the revised Introduction, the scope and method of this contribution is described as follows (line 54ff.):

“Against this background, it is of interest to examine to what extent scientific research originating from MICs has already addressed these latter challenges. Using a narrative literature review, this contribution explores the evolution of psychosocial occupational health research in recent past in two regions of MICs, Latin America and Asia Pacific. As occupational health research is quite developed in several countries within these regions there is reason to believe that the newly occurring problems of stressful psychosocial work have been tackled to some extent.”

As described in the Methods section (line 171ff), the narrative review has been extended by conducting a search strategy of major journals over the past few years, identifying some 80 new papers. The essence of this new review is summarized in the text of the Results section, illustrating some prominent papers in an extended list of references.

  1. The Methods section describes the journals and the time frame of the newly conducted narrative review. It justifies their selection and points to the important exclusion of non-English language publications.
  1. The Discussion section was extended to emphasize major limitations, including the lack of generalization beyond the regions under study (see line 356ff).
  1. A Conclusions section was added (see line 369ff).

Round 2

Reviewer 2 Report

Comments and Suggestions for Authors

Title of manuscript: Psychosocial occupational health - a priority for middle-income
countries?

Journal: Healthcare

Manuscript ID: healthcare-2680208

November 2023

The new version of the article entitled "Psychosocial occupational health - a priority for middle-income countries?"  with a new ID healthcare-2680208 to be published in this prestigious journal "Healthcare " has not only responded to all the aspects suggested in my previous revision which was revised in October 2023. Furthermore, this new version has substantially improved its initial version.

Therefore, my assessment to the editor is acceptance for publication.

Reviewer 3 Report

Comments and Suggestions for Authors

Dear authors, thank you! You have taken into account all the recommendations. The article may be recommended for publication.

Best wishes, reviewer